# Commutable Blood Materials from the Fixed-Cell Method for Performance Evaluation of Blood Glucose by a Glucose Meter

**DOI:** 10.3390/diagnostics14080799

**Published:** 2024-04-11

**Authors:** Napaporn Apiratmateekul, Jintana Nammoonnoy, Gerald J. Kost, Wanvisa Treebuphachatsakul

**Affiliations:** 1Department of Medical Technology, Faculty of Allied Health Sciences, Phitsanulok 65000, Thailand; napaporna@nu.ac.th; 2Reference Material and Laboratory Innovation Research Unit, Naresuan University, Phitsanulok 65000, Thailand; 3National Institute of Metrology (Thailand), Prathum Thani 12120, Thailand; jintana@nimt.or.th; 4Pathology and Laboratory Medicine, POCT.CTR, School of Medicine, University of California, Davis, CA 95616, USA; geraldkost@gmail.com

**Keywords:** equivalent result, exchangeability, proficiency testing, control material, reference material, matrix effect

## Abstract

Glucose meters provide a rapid blood glucose status for evidence-based diagnosis, monitoring, and treatment of diabetes mellitus. We aimed to evaluate the commutability of processed blood materials (PBMs) and their use in the performance evaluation of glucose meters. Two PBMs obtained by the fixed-cell method were analyzed for homogeneity, stability, and commutability. The compatibility of ten pairs between mass spectrometry and each glucose meter was categorized as compatible (mean paired difference ≤ 5%) and incompatible (mean paired difference > 5%). The performance of glucose meter 1 (*n* = 767) and glucose meter 2 (*n* = 266) was assessed. The glucose in the PBMs remained homogenized and stable for at least 180 days. Six out of ten pairs had commutable PBMs. Commutability of PBMs was observed in both well-compatible and incompatible glucose results. Target glucose values from mass spectrometry were significantly different (*p* ≤ 0.05) from consensus values in one group of glucose meters. When commutable PBMs were used, glucose meter 1 showed better performance than glucose meter 2, and the percentage of satisfaction was associated when using target values for glucose from mass spectrometry and consensus values, but the performance of glucose meter 2 was not associated. PBM from a fixed-cell method could be mass produced with acceptable homogeneity and stability. Commutability testing of PBMs is required prior to use in the performance evaluation of glucose meters, as the commutability of glucose in the PBMs obtained by a fixed-cell method was variable and depended on the individual glucose meter.

## 1. Introduction

Blood glucose measurements using glucose meters are widely used for point-of-care testing (POCT) diabetes management in hospitals and public health services [1,2,3]. Glucose meters provide a rapid blood glucose status that contributes to evidence-based diagnosis, monitoring, and treatment of diabetes mellitus in both small communities and large cities [4,5]. The implementation of internal quality control (IQC) and external quality assessment (EQA) is imperative for POCT. The IQC material for glucose meters comprises an aqueous control material provided by manufacturers specific to their devices. Adequate control samples in substantial quantities are essential for EQA to ensure the reliability of POCT. One of the primary challenges in EQA lies in the stability and commutability of blood materials, especially when dealing with a large number of participants. 

Commutability is a mathematical relationship between the results of different measurement methods for a processed sample and representative clinical samples (from healthy and sick individuals). Commutability, exchangeability, and interchangeability may also be defined as qualitative terms used to describe the similarity between processed blood materials (PBMs) and native clinical samples [6].

Although blood materials for glucose testing can be prepared from fresh human blood with heparin as an anticoagulant, there are some limitations due to cellular metabolism and difficulties in mass production. Alternatively, PBMs from donor blood that have undergone various processing methods are feasible for mass production. The use of human blood for PBMs is imperative because the matrix of the blood base resembles real human blood samples. The chemical or physical processes may affect the matrix of the materials and result in non-commutable glucose in PBMs. 

The use of non-commutable PBMs increases the mean difference between two different measurement principles compared to native clinical blood samples [7,8]. In addition to commutability, other properties of PBMs, such as homogeneity and stability, should also be considered according to the international guidelines for reference materials [9,10]. Commutable blood materials with stable glucose concentrations are essential for comparing the performance of different glucose meters. 

The objectives of this study were to evaluate the commutability of PBMs prepared by a fixed-cell method at two glucose concentrations and to use these commutable PBMs to evaluate the performance of glucose meters against glucose target values established through mass spectrometry and consensus.

## 2. Materials and Methods

### 2.1. Instruments 

This study utilized one mass spectrometer and five glucose meters, the most commonly used POCT instruments for glucose testing in Thailand. The professional metrologists of the National Institute of Metrology, Thailand (NIMT) set up the mass spectrometer technique for the measurement of glucose according to the modified published liquid chromatography–isotope dilution tandem mass spectrometry method (LC-ID-MS/MS) [11]. The measurement was performed using an LC-MS/MS system consisting of a Shimadzu system controller, LC -30AD pumps, a SIL-30AC autosampler, a CTO -20AC column oven, and an LCMS-8060 mass spectrometer (Shimadzu, Tokyo, Japan). The NIST SRM 917c (d-glucose) [12]. and NIST SRM 965b (glucose in frozen human serum) [13]. Standard reference materials were used to verify the instrument’s accuracy. 

All plasma samples were prepared gravimetrically. Sample mixtures were prepared by weighing 100 mg of the samples and adding an isotopically labeled glucose solution. A six-step calibration curve from 50 mg/dL to 250 mg/dL glucose with 75 mg/dL isotopically labeled glucose in each step was prepared gravimetrically. The weight of each solution was recorded to accurately calculate the mass ratios of glucose and isotopically labeled glucose. The mixture was shaken and allowed to stand for 1 h to obtain equilibrium. Next, 1 mL of acetonitrile was added for protein precipitation. The mixture was shaken and centrifuged at 4 °C and 12,000 rpm for 10 min. The supernatant was filtered through a 0.22 µm syringe filter before mass spectrometric analysis.

### 2.2. Native Blood Samples

Eight milliliters of native, unprocessed original venous blood samples were collected from 30 participants (10 healthy individuals and 20 patients with diabetes mellitus). Informed consent was taken from all participants included in this study. Plasma samples were stored at −21 °C and transported to the NIMT to measure glucose fractions using LC-ID-MS/MS. Samples were weighed on a Mettler Toledo XPE205 balance with a readability of 0.01 mg and a maximum capacity of 220 g (Mettler-Toledo Inc., Greifensee, Switzerland).

### 2.3. Processed Blood Material

The Research and Development Department of We Med Lab Center Co. Ltd. in Phitsanulok City, Thailand, has developed three PBMs, named BG62—001, BG62—002, and BG62-003, using the fixed blood cell method that was modified from US patent#4731330A [14].

For this purpose, a solution of glutaraldehyde (Sigma Aldrich, Tokyo, Japan) in phosphate-buffered saline (PBS) at a concentration of 0.75% was carefully prepared. The packed red blood cells, which were obtained from a blood bank, were thoroughly washed three times in 0.85% normal saline solution (NSS). The fixed solution, in which a precise ratio of 9:1 to the volume of red cells was maintained, was then applied and rolled in a tube for 2 h to facilitate fixation with glutaraldehyde.

After fixation, the cells were thoroughly washed in triplicate with PBS and then resuspended in PBS with a preservative solution. To evaluate cell stability, each sample, consisting of 3 milliliters, was transferred to a new tube and centrifuged at 3000 rpm for 5 min. The supernatant was then analyzed spectroscopically to determine the percentage of lysed cells based on hemoglobin content.

PBMs were prepared by combining fixed blood cells in PBS with a single plasma donor in a precise 2:1 ratio. The glucose concentration in the PBMs was carefully adjusted to three different values. The hematocrit of the PBMs was adjusted in the range of 35–45%. In addition, the pH of the blood materials was adjusted with PBS to a value between 7.35 and 7.45. The manufacturing facility has a quality management system that meets the requirements of ISO 13485: Medical Devices-Quality management systems-Requirements for regulatory purposes [15] and in accordance with ISO 17034: General Requirements for the Competence of Reference Material Manufacturers [9]. Next, 1 L of each PBM was aliquoted at two glucose concentrations at 0.5 mL per vial and stored at 25 °C for 1 week prior to the experiment. Storage conditions were monitored using a calibrated temperature and humidity data logger.

### 2.4. Determination of Glucose in Unprocessed Native Blood and PBMs

Glucose concentrations in unprocessed native venous blood samples and PBMs were measured using LC-ID-MS/MS at the NIMT and five glucose meters at the Medical Devices Research Laboratory at Naresuan University (Phitsanulok, Thailand).

### 2.5. Homogeneity of Glucose in PBMs

A homogeneity study was performed according to the procedures described in ISO Guide-35: Reference materials-Guidance to characterize and assess homogeneity and stability [10]. Fifteen vials were randomly selected from the 1800 vials of each PBM. Glucose concentration was determined using a glucose meter based on amperometry glucose dehydrogenase (GDH) in duplicates per vial.

### 2.6. Stability of Glucose in PBMs

The stability of glucose in the three PBMs was assessed using a classical approach. One hundred vials of each PBM were kept in a storage room at 28 ± 3 °C. Five vials of the PBMs were randomly selected, and glucose concentration was measured with a glucose meter based on amperometry GDH in duplicates per vial at 0, 30, 60, 90, 120, and 180 days. 

Two PBMs were additionally prepared using the fixed blood cell method to test long-term stability. Prediction intervals were utilized to predict shelf life for a linear trend of the three PBMs using the same stability test data set. Uncertainties of measurement for stability (us_ta_) were calculated [10]. 

### 2.7. Commutability Assessment 

The commutability of glucose in PBMs was investigated to determine whether the processed materials displayed similar behaviors to the unprocessed native blood samples [6]. This study measured glucose levels in the native blood samples, while the two measurement methods were used for PBMs and the native blood samples. The two PBMs, with different glucose concentrations and acceptable homogeneity and stability, were used for the commutability study. Native blood samples and PBMs were measured using five POCT glucose meters at the Medical Device Research Laboratory at Naresuan University, which is certified according to ISO 15189: 2012 [16].

### 2.8. Performance Evaluation of Blood Glucose Measurement by Glucose Meters Using PBMs

PBMs BG62—001 and BG62—002 were used as proficiency testing materials and further tested for homogeneity. Stability was monitored at the WE Med Lab Proficiency Testing Center before shipment to the participants, who were tested with glucose meter 1 and glucose meter 2 in the proficiency testing program. Glucose meter 1 with amperometry GDH-PQQ comprised 767 m, while glucose meter 2 with amperometry GDH-FAD comprised 266 m. The WE Med Lab Proficiency Testing Center collaborates with Naresuan University and is certified according to ISO /IEC 17043: 2010 [17]. 

### 2.9. Statistical Analysis

The Cochran test was used to detect outliers with 95% confidence intervals. Glucose homogeneity between samples in the 15 randomized vials was determined using the one-way analysis of variance (ANOVA) single factor according to ISO Guide-35 [10]. The glucose stability of PBMs was assessed by comparing the glucose concentrations from each day against the baseline (day 0). 

All glucose values were analyzed for commutability using the Deming regression plot according to CLSI-EP 14A [6]. Glucose values of the two PBMs were characterized using a mass spectrometer and five glucose meters as the mean and standard deviation. All assigned values were compared by the two-way ANOVA with Tukey's honestly significant difference (HSD) test. 

Target values from LC-ID-MS/MS and consensus (robust means and robust standard deviations [SDs]) [18] in one group of glucose meters were used in the performance evaluation of the two glucose meters. Satisfactory performance was assessed by mean differences according to ISO 15197: 2013 [19]. The mean differences were to be within 15 mg/dL when glucose levels were less than 100 mg/dL and within 15% when levels were greater than 100 mg/dL.

## 3. Results

### 3.1. Homogeneity of Glucose in PBMs

The homogeneity of glucose between samples in the two PBMs was measured using a glucose meter based on amperometry GDH. Glucose levels between samples were sufficiently homogeneous, as F_calculate_ was less than F_critical_ (Table 1). 

### 3.2. Stability of Glucose in PBMs

The stability of glucose in the PBMs was 180 days, as T_calculate_ was smaller than T_critical_ (Table 2), with the temperature in a storage room maintained at 29.0 ± 2.0 °C (range, 23–33 °C) and humidity at 70 ± 11% (range, 45–81%).

The prediction interval (PI) (Figure 1) showed that the glucose in the PBMs had long-term stability. The prediction lines intersected the lower or upper PIs at approximately 215 and 195 days for PBM BG62—001 and PBM BG62—002, respectively.

### 3.3. Commutability of Glucose

Commutability of the PBMs was observed in both compatible (A) and incompatible (B) glucose results. The characterization of glucose in the two PBMs and the mean and SD of glucose values in the PBMs are summarized in Table 3. The statistical significance of the glucose values in the two PBMs, characterized using five glucose meters and mass spectrometry, was determined by the two-way ANOVA with Tukey’s HSD test. The compatibility of glucose values in the two PBMs between each glucose meter and mass spectrometry is shown in Table 3. Compatible glucose results were obtained when the mean differences of the individual paired data between the glucose meter and mass spectrometry were <5% with *p* ≥ 0.05.

The commutability of glucose in the PBMs between the two data sets of each glucose meter and mass spectrometry was presented in parallel with the mean differences (Table 3). There was commutable glucose in PBM BG62—001 and PBM BG62—002.

The Deming regression plots (Figure 2A–E) represent the commutability of glucose in the PBMs between LC-ID-MS/MS and each POC glucose meter. Glucose in the PBMs was commutable between LC-ID-MS/MS and each glucose meter as it was in the region between the lower and upper PI. Commutable PBMs were identified in six out of ten pairs of glucose meters and mass spectrometry. The mean deviations of the commutable PBMs ranged from 4.6% to 25.7%. On the other hand, the mean deviations of the non-commutable PBMs ranged from 0.5% to 49.2% (Table 3).

### 3.4. Performance of Blood Glucose by Two Glucose Meters

Target glucose values using mass spectrometry were significantly different (*p* < 0.05) from consensus values in one group of glucose meters. The percentage of satisfaction (mean difference ≤ 15 mg/dL or ≤15%) when using two PBMs with stated commutability is summarized in Table 4. 

When commutable PBMs were used, the performance of glucose meter 1 agreed more closely with target values from mass spectrometry than glucose meter 2, and the percentage of satisfaction was associated when using consensus values, but the performance of glucose meter 2 was not associated.

## 4. Discussion

The use of a target value protocol is ideal for glucose meters in the EQA program; however, this assessment is challenging without stable and commutable blood materials. Glucose concentrations in PBMs are well-defined substances directly related to the amount of the substance in the International System of Units, in mmol/L, and can be characterized using mass spectrometry measurement techniques in a single laboratory. When using commutable materials (Table 4), the target values from mass spectrometry could be used in the performance evaluation of glucose meter 1 and glucose meter 2. 

The mean glucose values from the proficiency testing program (Table 3) differed slightly from those measured by expert laboratories (Table 4), depending on the number of laboratories in the proficiency testing program and the measurement competence of the participants at the proficiency testing program. The target value protocol for performance evaluation is the test glucose level deviation from the target values. In this study, 15 mg/dL, if glucose levels were below 100 mg/dL, or 15%, if levels were above 100 mg/dL, were used as decision criteria in the target protocol according to ISO 15197: 2013 [17]. When considering the performance of the two glucose meters, glucose meter 1 showed closer agreement to target mass spectrometry values than glucose meter 2; however, glucose meter 2 showed a better performance than glucose meter 1 when consensus values were used. Consensus values may not necessarily reflect accuracy or trueness. Nevertheless, when target values from mass spectrometry are used, differences in the glucose meters’ abilities can be assessed. The glucose values measured with glucose meter 2 were approximately 13.4–20.3% lower than those measured with mass spectrometry (Table 3) when performed from an expert laboratory, with the difference being greater with an increase in the number of users (Table 4). When used in clinical practice, glucose meter 2 may give values lower than the actual values, which may lead to ineffective blood glucose control, underdiagnosis when screening at-risk groups, or misdiagnosis of overt diabetes mellitus. 

A potential reason for the lack of association between the performance of glucose meter 2 and satisfaction levels when using target values for glucose from mass spectrometry and consensus values in this study could be errors in manufacturer calibration. The calibration hierarchy must be suitable for the purpose and have an acceptable uncertainty, and calibrators must not cause bias due to non-commutability. Glucose meter 1 performed better than glucose meter 2 when commutable PBMs were used, indicating that glucose meter 1 provides glucose values that are traceable to a common reference system. 

The commutability focuses on the difference between measured values in PBMs that does not exceed the limits of the difference between values measured in native blood samples (Figure 2). However, the comparison focuses on the mean value between the two measurement methods in the same PBM and subtracts the other value to obtain the mean difference. 

Homogeneity and stability tests were performed under indoor conditions (temperature 23–33 °C, relative humidity 45–81%) corresponding to the Thai climate and actual conditions in primary care units in Thailand [20,21]. The homogeneity test is designed to ensure that glucose differences between vials are sufficient when F_calculate_ is less than F_critical_.

The stability test of the glucose in the PBMs showed that the glucose had acceptable stability according to the LINEST statistic, and the values were within the values reported by the manufacturer of the reference material [10]. The predicted shelf life of the two PBMs was at least 195 days at room temperature and relative humidity with and without air conditioning during the experiments. The overall glucose stability in the PBMs was acceptable for quality control and assurance of blood glucose test applications by glucometers. The cost of transportation of quality control materials at a constant temperature, such as 4 degrees Celsius, is quite high, and there are limitations in service. Therefore, the stability of the glucose of two PBMs from a fixed cell method at temperatures in the range of 23–33 degrees Celsius is useful for transportation with cold gel packs to reduce costs for EQA program providers in Thailand.

The commercial aspect of this study is to ensure that the reference materials illustrate the performance of the blood glucose meters under real-life conditions. This is important for blood glucose meter manufacturers as it allows them to verify the accuracy and reliability of their products, which, in turn, increases customer confidence and satisfaction. Therefore, commutable reference materials can help manufacturers comply with regulatory requirements and standards and avoid costly recalls or legal issues. In addition, the benefit of using commutable reference materials is that they ensure the overall quality and competitiveness of glucose meters in the market. 

## 5. Conclusions

The performance of the different glucose meters was comparable when commutable PBMs were used. Commutability testing of PBMs is required prior to their use in the performance evaluation of glucose meters, as the commutability of glucose in the PBMs obtained by a fixed-cell method was variable and depended on individual glucose meters. Future studies should focus on factors that may influence the commutability of PBMs, such as the quality of blood raw materials and glucose concentration. 

## Figures and Tables

**Figure 1 diagnostics-14-00799-f001:**
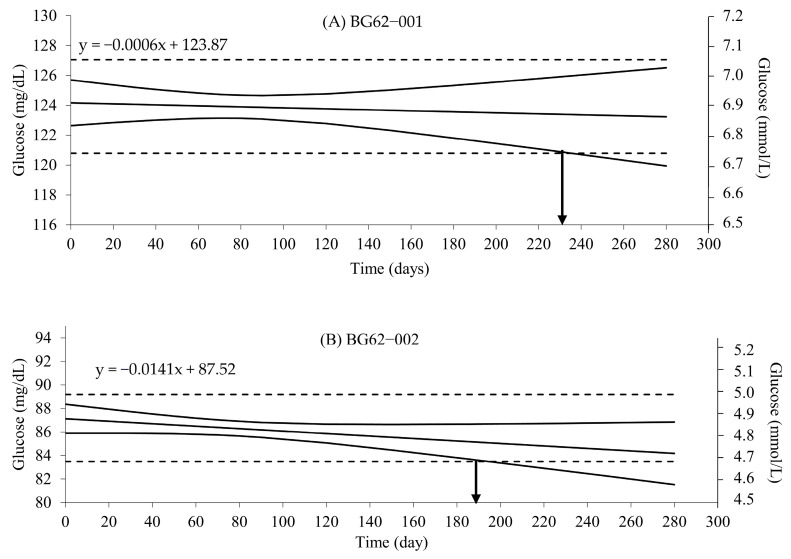
Shelf-life predictions of PBMs show that there was stabilization of glucose in two PBMs: (**A**) BG62—001 and (**B**) BG62—002. Long-term stability prediction is based on the accelerated stabilization of data provided. The 95% probability prediction interval is designated as the black dashed lines. Stability predictions were evaluated for two PBMs: (**A**) BG62—001 and (**B**) BG62—002. The solid blue lines represent the upper and lower acceptance criteria. The solid black line represents the prediction curve.

**Figure 2 diagnostics-14-00799-f002:**
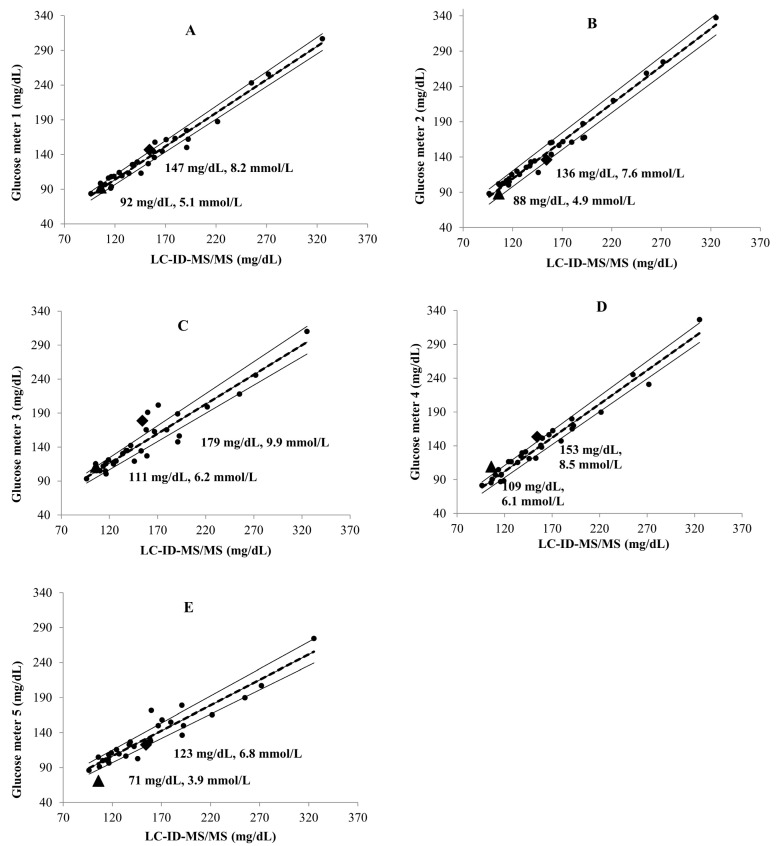
Deming regression for commutability of glucose in two PMBs: BG62—001 and BG62—002. (**A**) LC-ID-MS/MS vs. glucose meter 1; (**B**) LC-ID-MS/MS vs. glucose meter 2; (**C**) LC-ID-MS/MS vs. glucose meter 3; (**D**) LC-ID-MS/MS vs. glucose meter 4; (**E**) LC-ID-MS/MS vs. glucose meter 5. Dashed and solid lines represent the regression lines and the limits of the 95% PIs of Deming regressions, respectively. The black circles (●) represent the results of the PSs, and the black squares (♦) and black triangles (▲) represent the results of BG62—001 and BG62—002.

**Table 1 diagnostics-14-00799-t001:** Homogeneity of glucose in the two PBMs when mass production for 1800 vials was measured using a glucose meter based on an amperometry GDH.

PBMs	Mean (mg/dL)	CV (%)	F_calculate_	F_critical_	Interpretation
BG62—002	99	1.16	1.53	3.02	Adequate
BG62—001	135	1.33	2.28	3.02	Adequate

Processed blood materials (PBMs), glucose dehydrogenase (GDH).

**Table 2 diagnostics-14-00799-t002:** Stability of glucose in PBMs for 180 days at 29 ± 2 °C (range, 23–33 °C) with a relative humidity of 70 ± 11% (range, 45–81%) measured using a glucose meter with amperometry GDH.

Day	BG62—001			BG62—002		
	T_cal_	T_critical_	Interpretation	T_cal_	T_critical_	Interpretation
60	1.41	2.31	Acceptable	1.91	2.31	Acceptable
90	2.12	2.16	Acceptable	0.44	2.16	Acceptable
120	1.93	2.10	Acceptable	0.18	2.10	Acceptable
150	0.03	2.07	Acceptable	0.06	2.07	Acceptable
180	0.07	2.05	Acceptable	0.02	2.05	Acceptable

**Table 3 diagnostics-14-00799-t003:** Commutability of glucose in blood materials between values from mass spectrometry vs. each glucose meter and assigned values of glucose characterized from means and SD.

PBM	Mean ± SD (mg/dL)					
	LC-ID-MS/MS	Glucose Meter 1 Amp-GDH-PQQ	Glucose Meter 2 Amp-GDH-FAD	Glucose Meter 3 Amp-GOD	Glucose Meter 4 Pho-GOD	Glucose Meter 5 Amp-GDH-FAD
BG62—001	154 ± 0.9	C 147 ± 1.7 −7 (4.9%) A	C 136 ± 4.4 * −18 (13.4%) B	NC 179 ± 6.7 ** 25 (13.7%) B	NC 153 ± 7.8 −1 (0.5%) A	C 123 ± 3.2 ** −31 (25.7%) B
BG62—002	106 ± 1.4	C 92 ± 1.2 * −14 (14.7%) B	C 88 ± 2.7 * −18 (20.3%) B	C 111 ± 8.2 5 (4.6%) A	NC 109 ± 2.9 3 (3.1%) A	NC 71 ± 2.7 ** −35 (49.2%) B

Abbreviations: C, commutable material; NC, non-commutable material. A: Compatible when mean differences from mass spectrometry do not exceed 5%. B: Incompatible when mean differences from mass spectrometry exceed 5%. LC-ID-MS/MS, isotope dilution mass spectrometry; Amp-GOD, amperometry glucose oxidase method); Amp-GDH-PQQ, amperometry glucose dehydrogenase with the co-enzyme PQQ method; Amp-GDH-FAD, amperometry glucose dehydrogenase with the co-enzyme FAD method; Pho-GOD, photometry glucose oxidase method. * *p* < 0.05 is significant with a two-way ANOVA with Tukey’s HSD compared with mass spectrometry; ** *p* < 0.001 is significant with a two-way ANOVA with Tukey’s HSD compared with mass spectrometry.

**Table 4 diagnostics-14-00799-t004:** Acceptability from performance evaluations of blood glucose by two glucose meters of two commutable blood materials through the proficiency testing (PT) scheme using mean differences from mass spectrometry and consensus values as the target values.

Glucose Meter	1 Amp-GDH-PQQ (*n* = 767)	2 Amp-GDH-FAD (*n* = 266)
Range	79–157	113–138
Robust mean ± robust SD	141 ± 6.2 *	124 ± 4.5 **
	**Satisfaction**	
LC-ID-MS/MS Mean ± SD (154 ± 0.9 mg/dL)	761 (99.2%)	17 (6.4%)
Consensus values Robust mean ± robust SD	761 (99.2%)	266 (100.0%)
**Glucose Meter**	**1** **Amp-GDH-PQQ** **(*n* = 743)**	**2** **Amp-GDH-FAD** **(*n* = 246)**
Range	91–127	79–109
Robust mean ± robust SD	102 ± 3.6 *	90 ± 3.7 **
	**Satisfaction**	
LC-ID-MS/MS Mean ± SD (106 ± 1.4 mg/dL)	741 (99.7%)	112 (45.5%)
Consensus values Robust mean ± robust SD	739 (99.5%)	245 (99.6%)

* *p* < 0.05 is significant when compared with mass spectrometry by a *t*-test. ** *p* < 0.001 is significant when compared with mass spectrometry by a *t*-test. Satisfaction with mean difference according to ISO 15197: 2013 (the results should be within 15 mg/dL when glucose levels are less than 100 mg/dL and within 15% when levels are greater than 100 mg/dL).

## Data Availability

Data are available upon reasonable request from the author.

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
