# Peer review of "Commutable Blood Materials from the Fixed-Cell Method for Performance Evaluation of Blood Glucose by a Glucose Meter"

_diagnostics, 2024, doi:10.3390/diagnostics14080799_

Round 1
Reviewer 1 Report
Comments and Suggestions for Authors
Overall, this study provides valuable insights into the commutability of processed blood materials (PBM) for evaluating glucose meters. Here are some minor review comments for consideration:
1. Clarify the definition of "commutability" early on to aid readers' understanding, as it's a key concept in the study.
2. Consider mentioning the implications of the significant differences observed between target glucose values from mass spectrometry and consensus values for one group of glucose meters. How might this affect clinical practice or device calibration?
3. You are using the same blood samples or different blood samples to measure blood glucose levels in both Table 1 and Figure 1.
4. Elaborate on the factors contributing to the better performance of glucose meter 1 over glucose meter 2 when using commutable PBMs. Are there specific design features or calibration methods that could explain this difference?
5. Suggest potential reasons why the performance of glucose meter 2 was not associated with satisfaction levels when using target values for glucose from mass spectrometry and consensus values. Was there variability in the meter's readings or user interface issues?
6. Conclude with clear recommendations for future research or clinical practice, considering the variability of commutability observed in PBMs obtained by the fixed-cell method.
Addressing these points will enhance the clarity and completeness of your study's presentation and interpretation.
Author Response
- Clarify the definition of "commutability" early on to aid readers' understanding, as it's a key concept in the study.
- Move the definition of commutability early on the 2nd paragraph and add more definition (lines 44-48).
- Consider mentioning the implications of the significant differences observed between target glucose values from mass spectrometry and consensus values for one group of glucose meters. How might this affect clinical practice or device calibration?
- Add more content on discussion (lines 303-316)
- You are using the same blood samples or different blood samples to measure blood glucose levels in both Table 1 and Figure 1.
- We use different blood materials and have added a sentence to describe this in a section on methods (lines 142-143).
- Elaborate on the factors contributing to the better performance of glucose meter 1 over glucose meter 2 when using commutable PBMs. Are there specific design features or calibration methods that could explain this difference?
- Add more content for discussion (lines 303-316).
- Suggest potential reasons why the performance of glucose meter 2 was not associated with satisfaction levels when using target values for glucose from mass spectrometry and consensus values. Was there variability in the meter's readings or user interface issues?
- Add more content for discussion (lines 303-316).
- Conclude with clear recommendations for future research or clinical practice, considering the variability of commutability observed in PBMs obtained by the fixed-cell method.
- Re-write the conclusions (lines 340-342).
Reviewer 2 Report
Comments and Suggestions for Authors
Dear Authors,
The presented MS ´Commutable blood materials from fixed-cell method for performance evaluation of blood glucose by glucose meter´ is very informative and provides a details evaluation and comparison of PBM with two types of glucose meters.
I like to point out that the author forgot to mention the country and full address for the 1st affiliation.
A general formatting and sentence editing are required to check all over the MS.
The comparison of the 2 glucose meter is presented well however it would be nice if the conclusion and discussion mention the commercial aspect of the application of this study.
The concluding remarks are very short Author needs to further explain the findings and the future research directions.
The Figure 2 graphs are very small to see the points and bullets to identify and understand. The author needs to find some way for a better presentation.
Author Response
I like to point out that the author forgot to mention the country and full address for the 1st affiliation.
· City and country added (line 5).
A general formatting and sentence editing are required to check all over the MS.
· Carefully proofread with author's guidance and grammar editing by an author's English expert and SAGE author service (Certification).
The comparison of the 2-glucose meter is presented well however it would be nice if the conclusion and discussion mention the commercial aspect of the application of this study.
- The commercial aspect of this study are mentioned on discussion (Lines 337-344).
The concluding remarks are very short. Author needs to further explain the findings and the future research directions.
- Add some content on conclusions about the future research (lines 347-349).
The Figure 2 graphs are very small to see the points and bullets to identify and understand. The author needs to find some way for a better presentation.
- Improved font and symbol size for better visual inspections (Figure 2).